# ChatGPT-4o can serve as the second rater for data extraction in systematic reviews

**Mette Motzfeldt Jensen**[1,2]*, **Mathias Brix Danielsen**[1,2], **Johannes Riis**[1,2], **Karoline Assifuah Kristjansen**[3,4], **Stig Andersen**[1,2], **Yoshiro Okubo**[5], **Martin Grønbech Jørgensen**[1,2]

**1** Department of Geriatric Medicine, Aalborg University Hospital, Aalborg, Denmark, **2** Department of Clinical Medicine, Aalborg University, Aalborg, Denmark, **3** Department of Clinical Medicine, Aarhus University, Aarhus, Denmark, **4** Department of Plastic and Breast Surgery, Aalborg University Hospital, Aalborg, Denmark, **5** Falls, Balance and Injury Research Centre, Neuroscience Research Australia, University of New South Wales, Sydney, Australia

* Mette.Malene@rn.dk

## Abstract

### Background

Systematic reviews provide clarity of a bulk of evidence and support the transfer of knowledge from clinical trials to guidelines. Yet, they are time-consuming. Artificial intelligence (AI), like ChatGPT-4o, may streamline processes of data extraction, but its efficacy requires validation.

### Objective

This study aims to (1) evaluate the validity of ChatGPT-4o for data extraction compared to human reviewers, and (2) test the reproducibility of ChatGPT-4o's data extraction.

### Methods

We conducted a comparative study using papers from an ongoing systematic review on exercise to reduce fall risk. Data extracted by ChatGPT-4o were compared to a reference standard: data extracted by two independent human reviewers. The validity was assessed by categorizing the extracted data into five categories ranging from completely correct to false data. Reproducibility was evaluated by comparing data extracted in two separate sessions using different ChatGPT-4o accounts.

### Results

ChatGPT-4o extracted a total of 484 data points across 11 papers. The AI's data extraction was 92.4% accurate (95% CI: 89.5% to 94.5%) and produced false data in 5.2% of cases (95% CI: 3.4% to 7.4%). The reproducibility between the two sessions was high, with an overall agreement of 94.1%. Reproducibility decreased when information was not reported in the papers, with an agreement of 77.2%.

**Data Availability Statement:** All relevant data are within the paper and its Supporting Information files.

**Funding:** The author(s) received no specific funding for this work.

**Competing interests:** The authors have declared that no competing interests exist.

## Conclusion

Validity and reproducibility of ChatGPT-4o was high for data extraction for systematic reviews. ChatGPT-4o was qualified as a second reviewer for systematic reviews and showed potential for future advancements when summarizing data.

## Introduction

Systematic reviews synthesize all available research and represent the highest level of evidence, which is crucial for knowledge transfer and informing best practices and clinical guidelines. Conducting systematic reviews is thus important but time-consuming, typically taking between 6 to 12 months, with some reviews extending longer for complex or voluminous topics [1, 2].

Artificial intelligence (AI), such as large language models (LLMs) (e.g., ChatGPT-4o by OpenAI), shows promise for certain procedures in a systematic review like literature search, screening, data extraction, analysis, and quality assessment [1]. Recent literature has explored the potential of AI in research methodologies, particularly in article screening. Van Dijk et al. investigated AI for title and abstract screening and found it time-saving, efficient, and useful when applied correctly [3]. Feng et al. evaluated AI for automated literature searches, using human investigators as a reference standard, and found a high recall rate but concluded that human involvement is indispensable for selecting relevant studies [4]. Ghosh et al. (2024) employed an In-Context Learning framework to extract PICO elements from clinical trials, achieving state-of-the-art results [5].

LLMs have shown promise in automating complex tasks, such as extracting data from clinical trial documents. Data extraction traditionally require significant manual effort and expertise as it directly affects the accuracy of conclusions. Errors or incomplete data can lead to misinterpretations and skew clinical recommendations. Manual extraction is labor-intensive, time-consuming, and prone to error, which impacts reproducibility and complicates handling large datasets. As the volume of published research grows, AI-based models are increasingly relevant to automate extraction, reduce human errors, and improve consistency.

There is consensus that AI is a promising tool for conducting systematic reviews, but it faces challenges and limitations that necessitate human assessment and emphasize the need for validation [6–8].

A 2021 systematic review concluded that the benefits of AI for data extraction were unclear, with AI tools by then deemed insufficient and unreliable [6]. However, recently, Santos et al. (2023) concluded that AI as Machine Learning and Natural Language Processing models was efficient for systematic reviews and clinical guidelines but noted limitations that need addressing before full automation can ensure accuracy and efficiency [7]. In a recent study, Alyasiri et al. (2024) demonstrated that ChatGPT-4 shows promise in reference retrieval. However, the accuracy and reliability of AI-generated references remain a concern, as the model occasionally generates incorrect or fabricated citations. These findings underscore the importance of human oversight in AI-assisted research and provide context for evaluating ChatGPT-4's capabilities in data extraction for systematic reviews [9].

On May 13, 2024, OpenAI launched the latest and improved LLM ChatGPT-4o, capable of understanding figures and tables. Thus, our present study explores whether this enhanced version can replace a human for data extraction during a systematic review.

### Aim

This study aims to (1) evaluate the validity of ChatGPT-4o for data extraction in systematic reviews for randomized controlled trials (RCTs) compared to two independent human reviewers as the reference standard, and (2) test the reproducibility of ChatGPT-4o's data extraction.

## Methods

### Protocol

The protocol was registered on OSF in November 2023 and updated in May 2024 to specify the aim and validation methods and report the use of the latest version of ChatGPT, ChatGPT-4o (https://osf.io/8gn4p/).

### Design

To test the validity and reproducibility of ChatGPT-4o data extraction compared to traditional manual data extraction, we used papers from our ongoing systematic review on exercise to reduce fall risk currently being performed by the author group [10]. The manual data extraction were performed by two independent reviewers, and a third reviewer was involved to settle any disagreement. Consensus results of the manual data extraction from the ongoing source review were used as the reference standard in this study.

The validity of ChatGPT-4o's data extraction was tested by comparing the reference standard to two sessions of data extraction using Chat-GPT-4o. Each session of ChatGPT-4o's data extraction was performed independently by one of two authors (MMJ and MD) with separate ChatGPT-4o accounts. Data extraction in ChatGPT-4o was performed on 14th to 17th of May 2024 following the publication of the updated ChatGPT-4o version on the 13th of May 2024.

To test the reproducibility of ChatGPT-4o's data extraction, we compared the results of the two sessions.

An overview of the study design is shown in Fig 1.

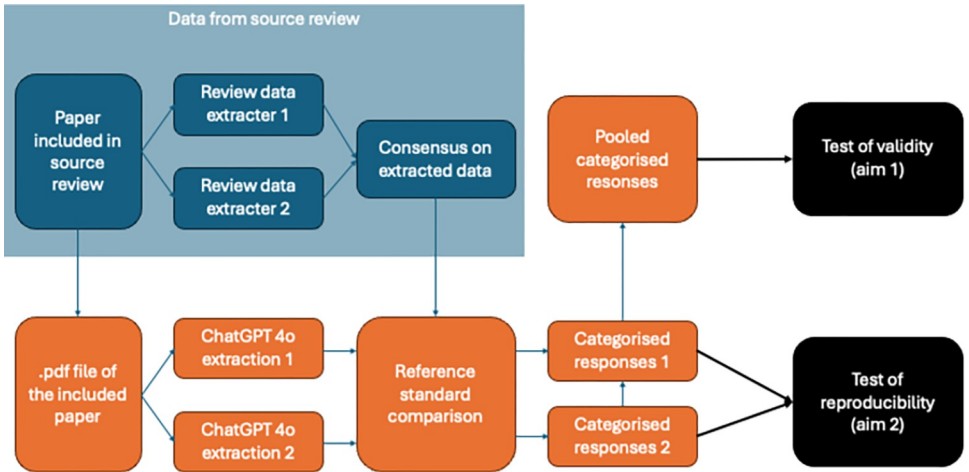

**Fig 1. Overview of the study design.** This study (orange) originates from a source review (blue). Articles included in the source review were used for data extraction with ChatGPT-4o. The 'Consensus on extracted data' from the source review was considered as the reference standard, against which ChatGPT-4o's responses were tested for validity and reproducibility (black).

**ChatGPT-4o data extraction.** Questions (prompts) for each data point were written without special knowledge of LLMs, phrased as they would be to another researcher. These questions covered baseline information, intervention description, participant baseline information, drop-out rates, and evaluation results for two outcomes (daily-life falls and laboratory falls). The questions were directly inserted as prompts in ChatGPT-4o. The list of prompts can be found in supplementary (S1 Table).

The relevant article PDFs included in the source review were uploaded to ChatGPT-4o. Two independent authors (MD and MMJ) entered all prompts into ChatGPT-4o and copied the responses into an Excel sheet. Only the first answer was used, with no regeneration or modification of answers. The authors used independent ChatGPT-4o accounts from different computers with the setting "Improve the model for everyone" turned off, and a new chat was used for each study. Turning the setting "Improve the model for everyone" off means that the input given to ChatGPT-4o is not used for model training and, therefore, prevents interference of the other data extraction session from the same paper.

**Reference standard.** Data extraction in the source review was conducted according to usual quality standards. Two authors extracted data from the included studies independently and afterwards reached consensus on the extracted data. The consensus data was used as a reference standard for the current study. Half of the included studies (n = 11) in the ongoing review were randomly selected to test ChatGPT-4o's ability compared to humans [10].

**Comparison of data extracted by ChatGPT 4o with the reference standard.** The data extracted by ChatGPT-4o was compared to the reference standard by the two independent authors who also entered the prompts into ChatGPT. The responses were assessed and categorized according to pre-specified categories. The categories and goals for validity were published in the OSF protocol (https://osf.io/8gn4p/). The categories were described as following:

*Category 1. Completely correct (only relevant information)*: to achieve this category, the results must fulfill the following criteria: a) correctly answer the question, b) no false information, c) no missing information, and d) no unnecessary information.

*Category 2. Satisfactory (all the relevant information, but also non-relevant information)*: to achieve this category, the results must fulfill the following criteria: a) correctly answer the question, b) no missing information, c) no false information on the question asked, and d) additional or unnecessary information is acceptable, even if false, as long as the answer to the question is correct.

*Category 3. Lacking information (some of the relevant information but not all)*: to achieve this category, the results must fulfill the following criteria: a) answers part of the question with correct information, b) missing some information, c) no false information on the question asked, and d) additional or unnecessary information is acceptable, even if false, as long as the answer to the question is correct.

*Category 4.* Missing all information (none of the relevant information): to achieve this category, the results must fulfill the following criteria: a) do not answer the question, and 2) do not provide false information on the question.

*Category 5.* False data (any amount of false information i.e. hallucinations): To achieve this category, the results must answer the question with false and misleading information.

**Validity goals and expected utility of ChatGPT as a data extraction tool.** Goals for acceptable validity and expected utility of ChatGPT as a reviewer were pre-defined, with categories ranging from "Single Reviewer" to "Useless" based on accuracy and proportion of false data, as stated in our protocol https://osf.io/8gn4p/. This assessment method was developed to ensure that the quality of the data extraction is as good or better than that of a single, second, or third reviewer. See Table 1.

**Table 1. Response assessment categories.**

| | |
|---|---|
| 100% correct<br>Answers Category 1 on all questions | Single reviewer |
| 80–99% correct<br>< 80% of correct answers are in category 2<br>< 10% false | Second Reviewer |
| 50–79% correct and<br>< 20% false. | Third Reviewer |
| Less than 50% correct<br>and/or > 20% false. | Useless |

Our pre-specified assessment of the validity of ChatGPT-4o depending on category and degree of correct and false answers (left), and consequent implementation of ChatGPT-4o (right).

## Statistical methods

Frequencies and percentages were used for categorical data descriptions. Validity was analyzed by collecting data extracted by both raters/authors for each question, allowing for non-deterministic algorithm performance. Proportions of each response category were visualized using histograms with 95% confidence intervals. Post hoc subgroup analyses were performed based on the presence or absence of information in the uploaded PDFs and the type of information.

Reproducibility was tested by comparing the stability of the response categorization for a data point between two data extractions. The agreement between the two data extractions was generally high, so the Kappa values were deemed unreliable due to the "Kappa paradox" [11]. Instead, we reported the percentage agreement as the primary measure together with Gwet's AC2, which is more reliable when the agreement is high [12]. This value functions like a Kappa ranging between 1 and 0, and values towards 1 are better. Quadratic weighting was used to penalize greater discrepancy between ratings.

All analyses were performed using R version 4.1.2.

## Results

We included 22 prompts (S1 Table) of which the response from ChatGPT-4o served as data points, to be extracted from each of the 11 papers in the source systematic review. This resulted in 242 data points extracted by each of the two authors comparing data from ChatGPT-4o directly with standardized forms of the reference standard. The data was carefully evaluated within one of the five response categories. Both datasets with evaluations are available as supplementary (S1, S2 Datasets). In the test of validity, these 242 data points extracted by each author were stacked for a total of 484 data points. Of the extracted data points, 48 (15.5%) were not reported in the studies, requiring ChatGPT to recognize this information as missing. The time to complete data extraction by ChatGPT was 3.5 hours within a week, while human raters took approximately 25–30 hours over 6–7 weeks (part-time).

### Agreement between AI and human data extraction

The overall number of data points categorized as "completely correct" or "satisfactory" was 447 (92.4%, 95% CI: 89.5% to 94.5%). The amount of extracted data that was completely false was 25 (5.2%, 95% CI: 3.4% to 7.4%). The results for each category are shown in Fig 2. Based on these results, ChatGPT meets the prespecified goal for validity required to function as a second reviewer.

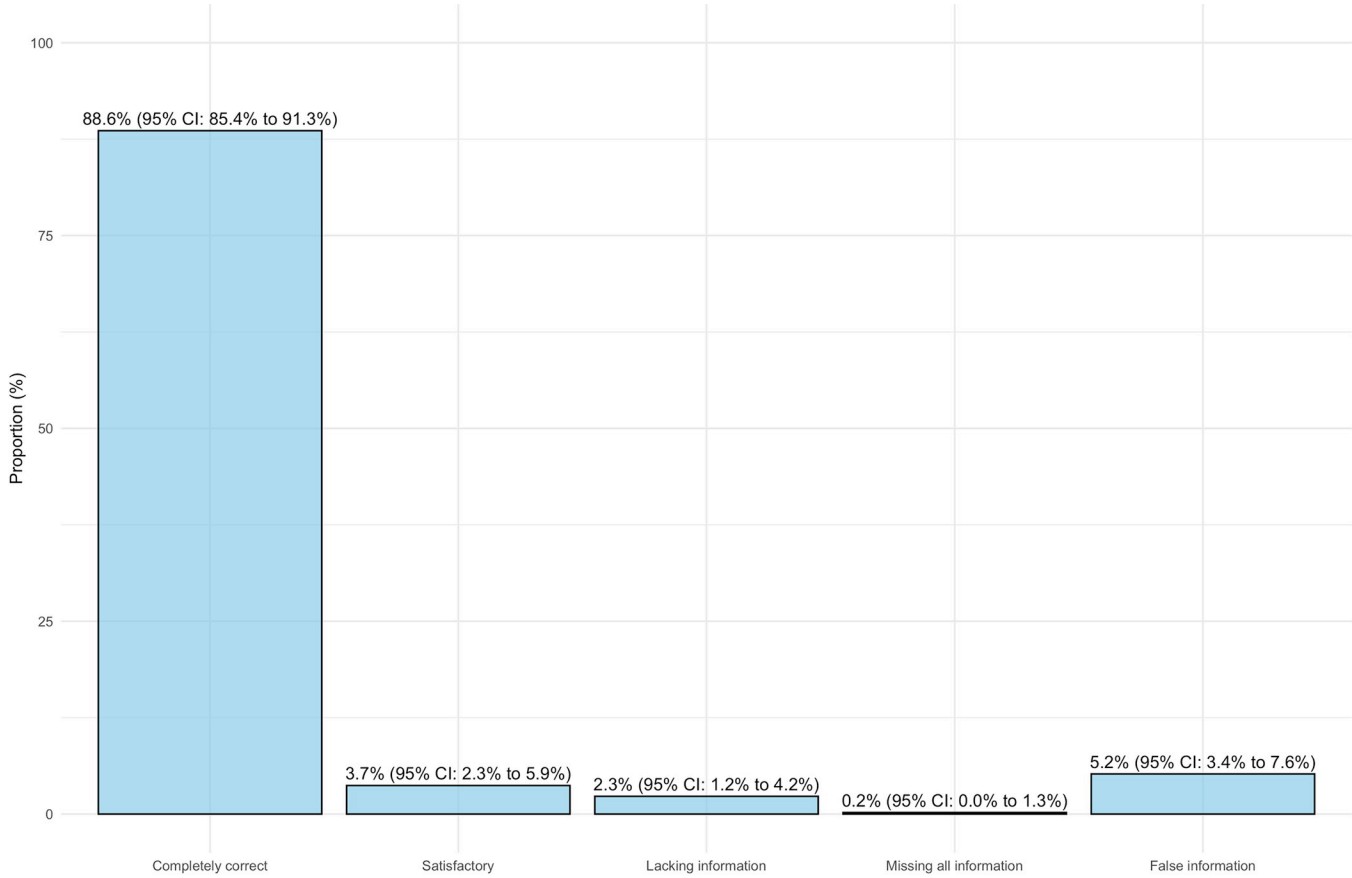

**Fig 2. ChatGPT-4o response assessment.** Evaluation of ChatGPT-4o responses compared to the reference standard. Response categories included 'Completely correct,' 'Satisfactory,' 'Lacking information,' 'Missing all information,' and 'False information'.

Results stratified by the presence or absence of information in the uploaded PDFs revealed a lower number of "completely correct" and "satisfactory" responses but higher frequencies of "false information" when data points were not reported in the study (Fig 3). Similar tendencies were found when dividing results based on the category of extracted data, with outcome data showing lower frequencies of completely correct or satisfactory information (Fig 4).

## Reproducibility of ChatGPT data extraction

The overall reproducibility between the two data extractions using ChatGPT was high, with an overall agreement of 94.1% and Gwet's AC2 of 0.93 (0.89 to 0.96) (Table 2). However, reproducibility was lower when the information was not reported in the paper, with agreements of 77.2% and Gwet's AC2 of 0.43 (0.10 to 0.75). This finding was further supported by the analysis of information domains, where agreement was lower for outcome data, and a larger proportion of information was not reported.

## Discussion

This study aimed to evaluate the validity of ChatGPT-4o for data extraction in systematic reviews for RCTs compared to a final consensus dataset established by two independent human reviewers and secondly to test its reproducibility. We found a 92.4% agreement between ChatGPT-4o and the human datasets, with only 5.2% completely false data points.

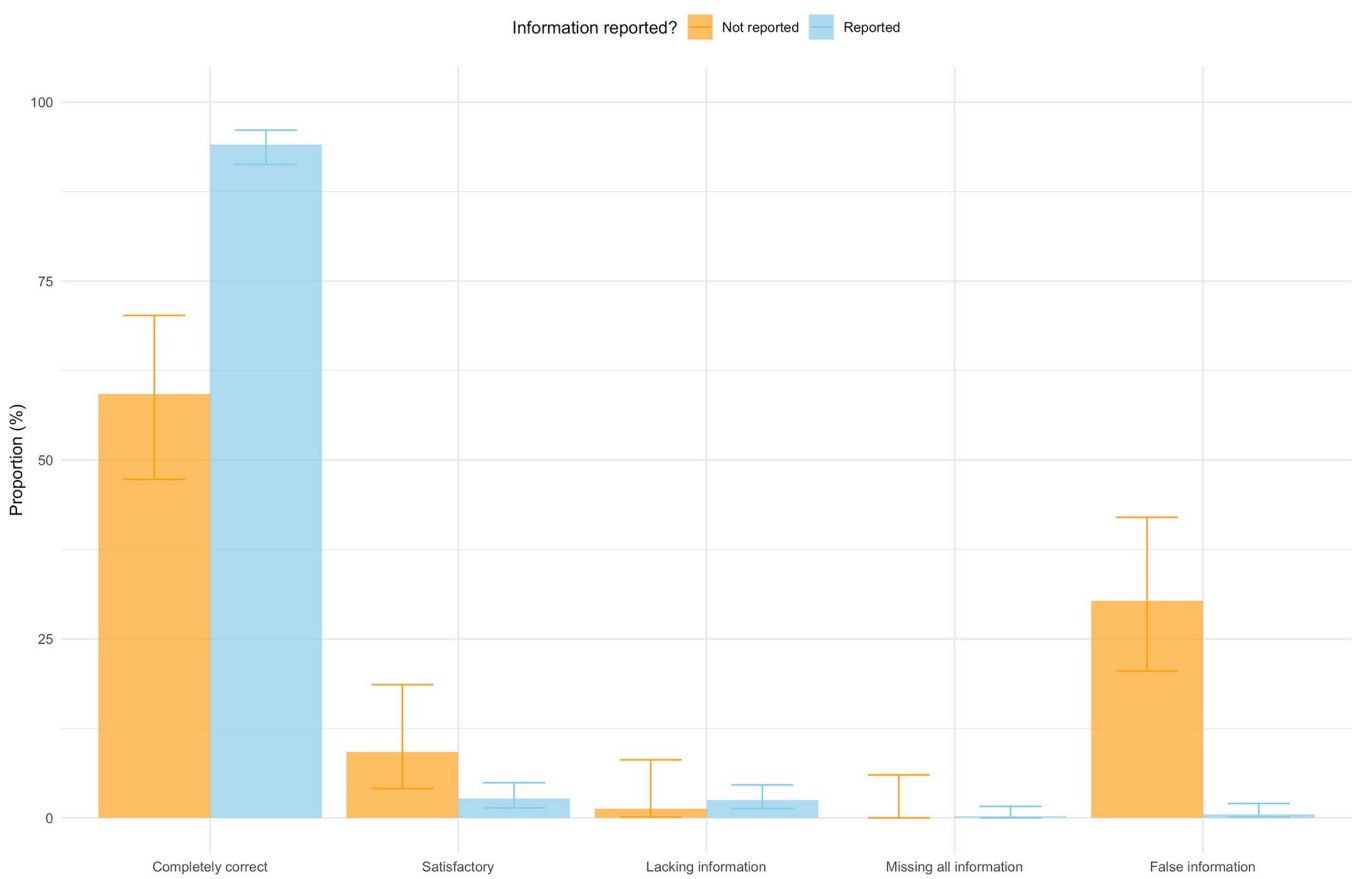

**Fig 3. Stratification of ChatGPT-4o responses by outcome reporting status.** The proportion of ChatGPT-4o responses in each response category stratified based on whether the articles reported the outcome or not.

Overall, reproducibility was high; however, when requested information was absent in the paper, the likelihood of false data was higher.

We are the first to explore the validity and reproducibility of ChatGPT-4o in data extraction for systematic reviews. Previous studies did not extensively examine data extraction using prior versions of ChatGPT. In a brief feature article, the use of ChatGPT for data extraction and Risk of Bias assessment was explored using the first public version of ChatGPT, ChatGPT 3.5, which was released by OpenAI in November 2022 [8]. The study found that while ChatGPT can aid in the Risk of Bias and data extraction process, it could not fully replace human expertise. The report also emphasized the importance of clear instructions for effective AI assistance and noted the limitations of ChatGPT-3.5. The updated ChatGPT4o has enhanced the ability to extract relevant information from large datasets, directly analyze uploaded PDFs, and interpret data from tables and figures within documents, making it a more robust tool for systematic reviews. In the present study, we show that while ChatGPT-4o cannot fully replace human expertise, it can serve as a second reviewer, reducing the workload and time required for a full researcher. Exploring the role of AI for data extraction is particularly relevant, as it also may address the existing inefficiencies and limitations of manual approaches. Compared to errors in human reviewers' data extraction (17.7% for a single rater and 14.5% for double raters), a mere 5.2% error was seen by the ChatGPT-4o, which qualifies it as an efficient second reviewer [13]. A recent study on the use of ChatGPT-3.5 Turbo in systematic reviews concluded the model may be used as a second reviewer for title and abstract screening [14].

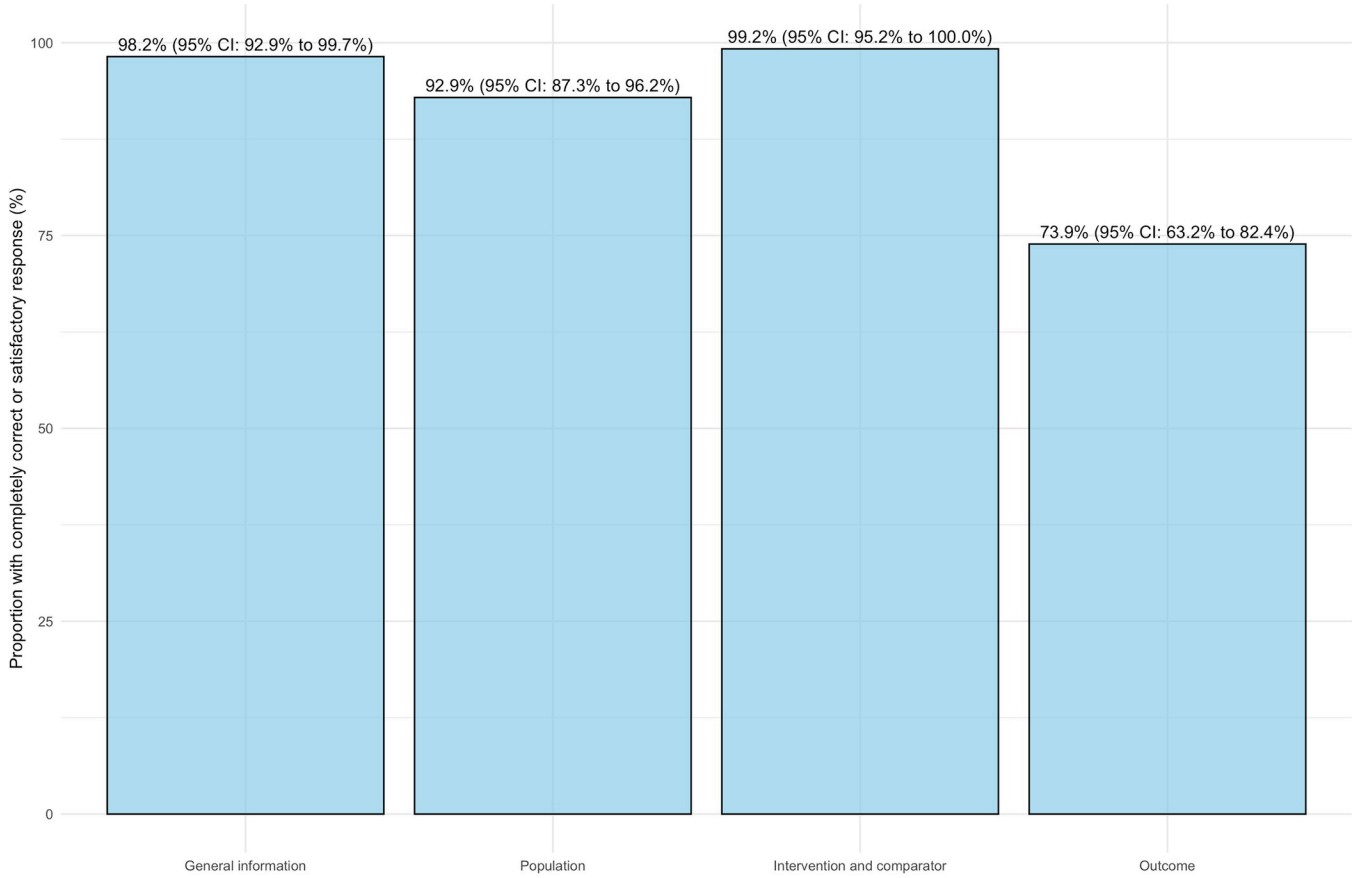

**Fig 4. Proportion of correct and satisfactory ChatGPT-4o responses across four data domains.** The proportion of 'Completely correct' and 'Satisfactory' responses from ChatGPT-4o compared to the reference standard across four data domains: General information, Population, Intervention and comparator, and Outcome.

**Table 2. Overall and domain-specific results of reproducibility of data extraction by ChatGPT-4o.**

| Domain | Percentage agreement | Gwet's AC2 (95% CI) |
|---|---|---|
| Overall results | | |
| All datapoints | 94.1% | 0.93 (0.89 to 0.96) |
| Data to extract reported vs. not reported in paper | | |
| Reported in paper | 98.2% | 0.98 (0.96 to 0.99) |
| Not reported in the paper | 77.2% | 0.43 (0.10 to 0.75) |
| Based on domain | | |
| General information | 98.3% | 0.98 (0.96 to 1.00) |
| Population | 95.4% | 0.94 (0.89 to 1.00) |
| Intervention and comparator | 99.3% | 0.99 (0.98 to 1.00) |
| Outcome | 78.7% | 0.60 (0.35 to 0.85) |

Results of reproducibility across all data points (top), including analyses based on whether data was reported or not in the paper, and by individual domains of extracted data (bottom).

While we demonstrated that ChatGPT-4o can serve as a second reviewer for data extraction, there is still room for improvement. A significant portion of the errors found in this study can be classified as hallucinations, where the model generated plausible but factually incorrect information, often occurring when specific data points were not explicitly reported in the articles. In such cases, instead of indicating missing information, ChatGPT-4o attempted to infer or generate data based on patterns from its training, leading to inaccuracies.

To overcome the limitation of incorrect data reporting, we prompted the AI to flag missing data rather than infer it. This was achieved by first asking whether the study collected the relevant data, followed by a request for specific details, e.g., 'Does the study collect data on laboratory falls? If yes, how is a laboratory fall defined, and how is information on laboratory falls collected?' Despite this approach, the model would still provide incorrect data or hallucinate. Further optimization or prompt engineering designed to ask more specific targeted questions or guide the model to recognize missing data, could be used to improve ChatGPT-4o's data extraction accuracy and reliability.

While we manually inserted the question in the prompt in the website version of ChatGPT-4o, taking 3.5 hours, this time can be drastically reduced by using a purpose-built plugin for ChatGPT-4o.

We have only explored the use of ChatGPT for data extraction, which serves as a significant advantage, as the data extraction process is among the most time-consuming steps in systematic reviews [13]. However, there is still a need to investigate the use of the latest version of ChatGPT or other similar AIs for other time-consuming aspects of systematic reviews, such as risk of bias assessment and abstract screening.

Our study is based on a source systematic review on RCT exploring training interventions for fall prevention in older adults. We assume ChatGPT-4o to produce similar results when used for data extraction in other fields. While this study demonstrates the potential of ChatGPT-4o as an auxiliary tool for data extraction in systematic reviews, it is important to acknowledge its limitations. The validity and reproducibility are markedly lower when the information is not reported in the primary article. Newer RCTs often follow a standardized reporting guideline to minimize missing information [15]. When ChatGPT-4o is applied for data extraction in other fields or other types of study, the risk of missing information must be considered, along with its potential to overlook complex contextual relationships. Although the model can handle large volumes of information, it may miss subtle interactions between study variables. Another concern is the presence of inherent biases raised from the data the model was trained on, resulting in overrepresentation of certain types of perspectives while underrepresenting others, and potentially skewing or misinterpreting the results of a systematic review. Future research should consider AI-based data extraction to be used alongside human reviewer. AI tools can assist in automating routine tasks and handling large datasets, while human expertise can focus on interpreting complex relationships and resolving ambiguous cases.

With an agreement between the two sessions of ChatGPT data extraction of 94.1%, the reproducibility was good; however, it was lower when the information was not reported in the paper, with an agreement of 77.2%. Furthermore, ChatGPT-4o performed best at answering general information about articles but faced challenges in the results section. Still, the tool is rapidly evolving, and the time required for data extraction is significantly shorter compared to human reviewers. Difficulties primarily arose when data was not reported, which could impact validity depending on the research area and quality of reporting.

## Ethical considerations in AI-driven systematic reviews

As AI models like ChatGPT-4o are increasingly integrated into research, ethical issues arise, particularly concerning accountability and transparency. While AI tools offer promising

advancements in automating data extraction and improving efficiency, researchers must regain control to ensure these technologies are used responsibly [16]. Hence, it is important to follow recent AI reporting guidelines [17].

It is essential that the role of AI is clearly disclosed in the research process. Accountability remains with the human researchers, who must oversee the AI's outputs and verify the accuracy of the extracted data. AI tools should be treated as complementary aids, with human reviewers ultimately responsible for validating the results and ensuring the overall quality of the review.

Addressing ethical issues will be critical as AI tools become prevalent in research, ensuring their use enhances, rather than undermines, the integrity of systematic reviews.

## Strengths and limitations

The study is based on only 11 RCTs on falls research, potentially limiting generalizability. While the sample size is relatively small and lacks stratification by study type, we ensured a robust methodology by including a large number of data points (total of 484 data points) which allows for a meaningful comparison with ChatGPT-4o. Expanding the study to include a broader range of topics or study types would however improve the external validity of our findings. Future research should explore ChatGPT-4o's performance across a wider variety of study designs. Incorporating more diverse data sources in future studies would help further assess the generalizability of ChatGPT-4o's performance and broaden its application across a wider range of systematic reviews.

ChatGPT-4o's responses were compared to a manual consensus reference standard, which is naturally subject to some degree of interpretation by the reviewers. Consequently, the evaluation of ChatGPT-4o's accuracy depends on how closely its responses align with the human reviewers' consensus, rather than an entirely objective measure. However, it is important to recognize that subjective variability is a common factor in data extraction for systematic reviews, even when performed solely by human reviewers.

The majority of the data extraction items in our study—such as mean age, gender distribution, and the number of dropouts—are based on objective information with little room for subjective influence, as there is typically only one correct answer for these data points. While we acknowledge that some level of subjectivity may persist, this approach is consistent with standard procedures in systematic reviews, where consensus-based human extraction is commonly used as the reference standard.

Finally, the authors do not have specific expertise in LLMs, and maybe better-written prompts could improve results. However, our approach exemplifies the average researcher's knowledge rather than the inner workings of LLMs, providing real-world results on the usability for the general researcher.

## Conclusion

We found the agreement between ChatGPT-4o and human reviewers to be greater than 80% for data extraction in systematic reviews, with a 92.4% agreement between ChatGPT-4o and human reviewers. We found a 5.2% error of data extracted by ChatGPT-4o. Hence, ChatGPT-4o can be used as a second reviewer in the data extraction process for systematic reviews. Large language models have developed rapidly in recent years and hold great promise for summarizing data to support clinical decision making as illustrated by the present evaluation of the advancement and validity of ChatGPT-4o. The performance may improve but it remains important to retain control of the process and improvements by further studies monitoring accuracy and reliability of LLMs.

## Supporting information

**S1 Table. List of prompts and domains.**
(DOCX)

**S1 Dataset. Data extraction from reviewer 1.** Full data extraction in ChatGPT-4o and evaluation of responses compared to the reference standard.
(XLSX)

**S2 Dataset. Data extraction from reviewer 2.** Full data extraction in ChatGPT-4o and evaluation of responses compared to the reference standard.
(XLSX)

## Author Contributions

**Conceptualization:** Martin Grønbech Jørgensen.

**Data curation:** Mathias Brix Danielsen, Stig Andersen, Yoshiro Okubo, Martin Grønbech Jørgensen.

**Formal analysis:** Mathias Brix Danielsen, Johannes Riis.

**Investigation:** Mette Motzfeldt Jensen, Johannes Riis, Karoline Assifuah Kristjansen, Stig Andersen, Yoshiro Okubo.

**Methodology:** Mette Motzfeldt Jensen, Mathias Brix Danielsen, Johannes Riis, Karoline Assifuah Kristjansen.

**Project administration:** Mette Motzfeldt Jensen, Mathias Brix Danielsen, Karoline Assifuah Kristjansen.

**Software:** Mathias Brix Danielsen.

**Supervision:** Martin Grønbech Jørgensen.

**Validation:** Mathias Brix Danielsen, Johannes Riis.

**Visualization:** Johannes Riis.

**Writing – original draft:** Mette Motzfeldt Jensen, Mathias Brix Danielsen, Johannes Riis, Karoline Assifuah Kristjansen.

**Writing – review & editing:** Mette Motzfeldt Jensen, Mathias Brix Danielsen, Johannes Riis, Karoline Assifuah Kristjansen, Stig Andersen, Yoshiro Okubo, Martin Grønbech Jørgensen.

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
