## [Decision Letter · Decision Letter 0]

29 Aug 2024

PONE-D-24-28092ChatGPT-4o Can Serve as the Second Rater for Data Extraction in Systematic ReviewsPLOS ONE

Dear Dr. Motzfeldt Jensen,

Thank you for submitting your manuscript to PLOS ONE. After careful consideration, we feel that it has merit but does not fully meet PLOS ONE’s publication criteria as it currently stands. Therefore, we invite you to submit a revised version of the manuscript that addresses the points raised during the review process.

**ACADEMIC EDITOR: After going through the manuscript and the reviewers' comments, the most reviewers and I suggest a Minor Revision,**
**Please review the following  details.**

We look forward to receiving your revised manuscript.

Kind regards,

Weiqiang (Albert) Jin, Ph.D.

Academic Editor

PLOS ONE

Additional Editor Comments:

Dear authors:

After going through the manuscript and the reviewers' comments, I suggest a Minor Revision (as a associate editor).

The study is promising and offers some great insights into using AI / gpt-4o for data extraction in systematic reviews, but there are a few areas that need a bit more work. The key points to address include adding more details about the methods—like how the sample size was chosen, the specific prompts used, and how the human reviewers did their work. Also, expanding the results and discussion sections to clarify the types of errors found, ways to reduce them, and any potential limitations would help. It's also important to make sure the manuscript follows the proper guidelines for reporting AI research and to polish the language for better readability.

Please consider all the suggestions given by the four reviewers and revise it, and provide us with a one by one detailed response letter.

regards,

Weiqiang Jin.

Reviewers' comments:

Reviewer's Responses to Questions

**Comments to the Author**

1. Is the manuscript technically sound, and do the data support the conclusions?

Reviewer #1: Yes

Reviewer #2: Partly

Reviewer #3: Yes

Reviewer #4: Yes

2. Has the statistical analysis been performed appropriately and rigorously? 

Reviewer #1: Yes

Reviewer #2: Yes

Reviewer #3: Yes

Reviewer #4: Yes

3. Have the authors made all data underlying the findings in their manuscript fully available?

Reviewer #1: Yes

Reviewer #2: Yes

Reviewer #3: Yes

Reviewer #4: Yes

4. Is the manuscript presented in an intelligible fashion and written in standard English?

Reviewer #1: Yes

Reviewer #2: Yes

Reviewer #3: Yes

Reviewer #4: Yes

5. Review Comments to the Author

Reviewer #1: Dear Editor and Authors

Thank you for submitting your manuscript. I have carefully read through the entire paper and believe this study presents an innovative approach with notable potential for enhancing data extraction processes in systematic reviews. The experimental design is robust, and the results are promising.However, there are a few areas where the manuscript could benefit from minor revisions to further improve clarity and impact.In summary, I recommend minor revision.Below are my specific comments, which I hope will assist you in revising and improving the paper.

1.Insufficient Detail in Background Description: The background section mentions that systematic reviews are a key tool for translating clinical trial evidence into guidelines but does not sufficiently explain why data extraction is a crucial step in systematic reviews and how it impacts the quality of the review. It is recommended to add a discussion on the limitations of current manual data extraction methods and the potential benefits of AI intervention in this section.

2.Results Section Needs More Comparative Detail: The results section compares data extraction by ChatGPT-4o with that of human reviewers but does not provide detailed information about the data extraction process used by human reviewers. To enhance the persuasiveness of the comparison, it is suggested to include detailed information about the background and data extraction criteria of human reviewers to ensure fairness and accuracy. Additionally, a description of the human reviewers' workflow could help better understand the differences and advantages between AI and human review processes.

3.Emphasis on Study Limitations in Discussion: While the conclusion affirms the potential of ChatGPT-4o as an auxiliary tool for systematic reviews, the study does not address its limitations, such as the potential for AI to overlook complex contextual relationships or inherent biases. It is recommended to discuss these potential limitations in the discussion section and explore ways to address these issues in future research.

4.Accuracy of Keywords: The keywords cover the core concepts of the study, but it may be beneficial to include more targeted terms such as “automated data extraction” and “machine learning” to enhance search relevance.

5.Data Sources and Representativeness: The study mentions randomly selecting 11 articles to test the capabilities of ChatGPT-4o but does not elaborate on the specifics of the random selection method or the representativeness of the sample. It is recommended to provide more information on how the sample was ensured to be representative, such as whether different types of studies or data reporting quality were considered, to ensure the generalizability of the test results.

Incorporating these adjustments will enhance the manuscript's overall quality and provide readers with a clearer understanding of the study's implications and applications.

Thank you for considering these suggestions.

Best regards,

Reviewer #2: This manuscript contributes valuable insights into the use of AI, specifically ChatGPT-4o, for data extraction in systematic reviews. However, there are some points that need further clarification and refinement.

1. External Validity:

1.1 The study's focus on a single ongoing systematic review raises concerns about the generalizability of the findings. Systematic reviews can encompass a wide range of research questions, including interventional, diagnostic, and prognostic studies, etc. It would be beneficial to consider whether the inclusion of different types of studies or systematic reviews might improve the validity and reproducibility of ChatGPT-4o's data extraction capabilities.

1.2 The sample size of 11 studies in one systematic review used to evaluate validity is relatively small. This limited sample size may not be sufficient to draw robust conclusions about the general applicability of ChatGPT-4o. Further justification for the adequacy of this sample size would strengthen the manuscript.

2. Introduction:

2.1 The Introduction would benefit from a more comprehensive review of existing literature on the use of large language models (LLMs) for data extraction. For example, including references such as DOI:10.1016/j.ymeth.2024.04.005, among others, would help contextualize the novelty and contributions of the current study.

3. Methods:

3.1 ChatGPT-4o data extraction：

The manuscript lacks details on the specific prompts used for data extraction with ChatGPT-4o. Providing this information would enhance the transparency and reproducibility of the study.

3.2 Comparison of data extracted by ChatGPT 4o with the reference standard:

It is unclear how the comparison between ChatGPT-4o and human data extraction was conducted. Specifically, were the evaluations conducted by one or two researchers? Clarifying this aspect of the methodology is important for assessing the rigor of the study.

3.3 Validity goals and expected utility of ChatGPT as a data extraction tool:

Providing a clearer reference to the categories for validity assessment would improve the clarity of the methods.

3.4 The manuscript mentions 22 data points extracted from each study but does not specify what these data points are. Detailing the specific data points would aid readers in understanding the scope of the data extraction process.

4. Results:

4.1 Consider providing examples of the typical data extraction results in the appendix. This would allow readers to better assess the performance of ChatGPT-4o.

4.2 The manuscript mentions that 5.2% of the data extracted by ChatGPT-4o was incorrect. A brief discussion of the types of hallucinations observed and potential strategies for identifying or mitigating these errors would be useful.

Other Comments:

5. Adherence to Reporting Guidelines:

The manuscript should align with reporting guidelines for AI-related clinical research, such as those mentioned in Flanagin et al. (2024), "Reporting Use of AI in Research and Scholarly Publication-JAMA Network Guidance." Providing details on the prompts used, the time frame of ChatGPT-4o usage, and other relevant methodological aspects would be beneficial.

6. Language and Clarity:

The manuscript contains some language that could be further refined for clarity. For example, the sentence "this finding was supported when looking across information domains where agreement was lower for outcome data and a larger proportion of information was not reported" could be simplified for better readability.

Reviewer #3: This is an interesting and novel study exploring the use of AI, LLM specifically, in collecting data for systematic reviews. However, I have a few comments that I would like the authors of this study to address.

1. Please provide how you arrived that a sample size of 11 would be adequate to test the validity and reproducibility of ChatGPT 4o.

2. I would be interested to know the agreement rate between the two authors? Suppose there was frequent disagreement between the two assessors, requiring a third author to intervene. Does this mean the data extracted by human standards was subjective and hence assessing ChatGPT against this subjective measure would have not shown its true capacity for extracting data from full-text papers?

3. How did you arrive at the prespecified assessment of the validity? Were there any previous similar studies using this model?

4. What was the prespecified goal of determining acceptable reproducibility?

5. All figures and tables should have a legend summarising and explaining their results. All figures and tables from this study are missing legends. Graphs are also missing titles.

Reviewer #4: 1. The study notes that the questions were written without special knowledge of LLMs, which reflects an average user’s experience. However, exploring how optimized prompts could enhance AI performance might provide valuable insights. This could lead to recommendations on best practices for researchers using ChatGPT-4o.

2. The study mentions that 5.2% of the data extracted by ChatGPT-4o was false, particularly when information was not reported in the papers. A deeper analysis of these errors—beyond reporting their frequency—could provide useful guidance on how to mitigate such risks.

3. While the manuscript discusses the validity and reproducibility of ChatGPT-4o, it might benefit from a brief discussion on the ethical implications of using AI in systematic reviews, particularly regarding accountability and transparency when AI-generated data is integrated into research outputs.

4. The study is based on a specific set of 11 RCTs focusing on fall prevention in older adults. While the methodology is robust, the generalizability to other fields or types of studies remains uncertain. What are the author's suggestions to expand the study to include a broader range of topics or study types that would strengthen the conclusions?

5. To add relevant information that supports the manuscript and benefits the readers, a suggested paper for investigating how ChatGPT can find and return real references. You may cite it as follows: https://doi.org/10.1016/j.jormas.2024.101842.

6. PLOS authors have the option to publish the peer review history of their article (what does this mean?). If published, this will include your full peer review and any attached files.

Reviewer #1: No

Reviewer #2: **Yes: **Suodi zhai

Reviewer #3: No

Reviewer #4: No

---

## [Author Response · Author response to Decision Letter 0]

3 Oct 2024

Dear editor and reviewers, 

I would like to sincerely thank you for your time and effort in reviewing our manuscript. We appreciate your valuable feedback and thoughtful suggestions, which has helped to provide more clarity and refining of our manuscript. We hope that the revised version meets your expectations, and we look forward to your further assessment.

Sincerely, 

Mette Motzfeldt Jensen

On behalf of the co-authors.

---

## [Decision Letter · Decision Letter 1]

24 Oct 2024

ChatGPT-4o Can Serve as the Second Rater for Data Extraction in Systematic Reviews

PONE-D-24-28092R1

Dear Dr. Authors of Paper PONE-D-24-28092R1,

We’re pleased to inform you that your manuscript has been judged scientifically suitable for publication and will be formally accepted for publication once it meets all outstanding technical requirements.

Kind regards,

Weiqiang (Albert) Jin, Ph.D.

Academic Editor

PLOS ONE

Additional Editor Comments (optional):

Congratulations! The reviewers have expressed their appreciation for your work and have acknowledged its quality by recommending acceptance of your article. Well done.

**Before the final proofreading, please ensure that all citations in the manuscript adhere to the publication's formatting guidelines.  Additionally, verify the accuracy of information for each referenced article, prioritizing published dois over preprints like arXiv.**

Reviewers' comments:

Reviewer's Responses to Questions

**Comments to the Author**:

I recommend citing the following two references that utilize GPT for information processing:

ChatAgri: Exploring potentials of ChatGPT on cross-linguistic agricultural text classification [DOI: 10.1016/j.neucom.2023.126708]

Prompt learning for metonymy resolution: Enhancing performance with internal prior knowledge of pre-trained language models [DOI: 10.1016/j.knosys.2023.110928]

Reviewer #1: All comments have been addressed

Reviewer #4: All comments have been addressed

2. Is the manuscript technically sound, and do the data support the conclusions?

Reviewer #1: Yes

Reviewer #4: Yes

3. Has the statistical analysis been performed appropriately and rigorously? 

Reviewer #1: Yes

Reviewer #4: Yes

4. Have the authors made all data underlying the findings in their manuscript fully available?

Reviewer #1: Yes

Reviewer #4: Yes

5. Is the manuscript presented in an intelligible fashion and written in standard English?

Reviewer #1: Yes

Reviewer #4: Yes

6. Review Comments to the Author

Reviewer #1: (No Response)

Reviewer #4: Thanks for addressed the comments raised in a previous round of review. So I feel that this manuscript is now acceptable for publication.

7. PLOS authors have the option to publish the peer review history of their article (what does this mean?). If published, this will include your full peer review and any attached files.

Reviewer #1: No

Reviewer #4: No

---

## [Editor Report · Acceptance letter]

5 Nov 2024

PONE-D-24-28092R1 

PLOS ONE

Dear Dr. Motzfeldt Jensen, 

I'm pleased to inform you that your manuscript has been deemed suitable for publication in PLOS ONE. Congratulations! Your manuscript is now being handed over to our production team.

Kind regards, 

on behalf of

Dr. Weiqiang (Albert) Jin 

Academic Editor

PLOS ONE